# Revisiting Topic-Guided Language Models

**Carolina Zheng**                                                          *carozheng@cs.columbia.edu*
*Department of Computer Science*
*Columbia University*

**Keyon Vafa**                                                                *kv2294@columbia.edu*
*Department of Computer Science*
*Columbia University*

**David M. Blei**                                                            *david.blei@columbia.edu*
*Department of Statistics*
*Department of Computer Science*
*Columbia University*

**Reviewed on OpenReview:** *https://openreview.net/forum?id=lXBEwFfxpA*

## Abstract

A recent line of work in natural language processing has aimed to combine language models and topic models. These *topic-guided language models* augment neural language models with topic models, unsupervised learning methods that can discover document-level patterns of word use. This paper compares the effectiveness of these methods in a standardized setting. We study four topic-guided language models and two baselines, evaluating the held-out predictive performance of each model on four corpora. Surprisingly, we find that *none of these methods outperform a standard LSTM language model baseline*, and most fail to learn good topics. Further, we train a probe of the neural language model that shows that the baseline's hidden states already encode topic information. We make public all code used for this study.

## 1 Introduction

Recurrent neural networks (RNNs) and LSTMs have been an important class of models in the development of methods for many tasks in natural language processing, including machine translation, summarization, and speech recognition. One of the most successful applications of these models is in language modeling, where they are effective at modeling small text corpora. Even with the advent of transformer-based language models, RNNs and LSTMs can outperform non-pretrained transformers on various small datasets (Melis et al., 2020).

While powerful, RNN- and LSTM-based models struggle to capture long-range dependencies in their context history (Bai et al., 2018; Sankar et al., 2019). Additionally, they are not designed to learn interpretable structure in a corpus of documents. To this end, multiple researchers have proposed adapting these models by incorporating topic models (Dieng et al., 2017; Lau et al., 2017; Rezaee & Ferraro, 2020; Guo et al., 2020). The motivation for combining language models and topic models is to decouple local syntactic structure, which can be modeled by a language model, from document-level semantic concepts, which can be captured by a topic model (Khandelwal et al., 2018; O'Connor & Andreas, 2021). The topic model component is also designed to uncover latent structure in documents.

We refer to these models as *topic-guided language models*. Broadly, this body of research has reported good results: topic-guided language models improve next-word predictive performance and learn interpretable topics.

In this work, we re-investigate this class of models by evaluating four representative topic-guided language model (TGLM) papers in a unified setting. We train the models from Dieng et al. (2017); Lau et al. (2017);

Rezaee & Ferraro (2020); Guo et al. (2020) on three document-level corpora and evaluate their held-out perplexity. Unlike some prior work, during next-word prediction, we take care to condition the topic model component on only previous words, rather than the entire document. Moreover, we use a baseline language model that is conditioned on *all* previously seen document words, rather than being restricted to the current sentence (Lau et al., 2017; Rezaee & Ferraro, 2020; Guo et al., 2020). Additionally, we choose baseline language models with comparable model sizes to ensure valid comparisons. Our finding: no predictive improvement of TGLMs over a standard LSTM-LM baseline (Zaremba et al., 2014).

In order to understand why topic-guided language models offer no predictive improvement, we probe the LSTM-LM's hidden representations. A probe is a trained predictor used to measure the extent to which fitted "black-box" models, such as neural models, have learned specific linguistic features of the input (Hewitt & Liang, 2019). The probe reveals that the LSTM-LM already encodes topic information, rendering a formal topic model component redundant.

Additionally, topic-guided language models were developed to provide insight into text corpora by uncovering latent topics. This method of exploratory text analysis is commonly used in the social sciences and digital humanities (Griffiths & Steyvers, 2004; Blei & Lafferty, 2007; Grimmer & Stewart, 2013; Mohr & Bogdanov, 2013). Here, we show that the topics learned by topic-guided language models are not better than a standard topic model and, for some of the models, qualitatively poor.

This paper contributes to a line of reproducibility studies in machine learning that aim to evaluate competing methods in a consistent and equitable manner. These studies have uncovered instances where results are not directly comparable, as reported numbers are borrowed from prior works that used different experimental settings (Marie et al., 2021; Hoyle et al., 2021). Furthermore, they identify cases where baselines are either too weak or improperly tuned (Dacrema et al., 2019; Nityasya et al., 2023). We observe analogous issues within the topic-guided language modeling literature. To support transparency and reproducibility, we make public all code used in this study.[1]

Finally, we consider how these insights apply to other models. While prior work has incorporated topic model components into RNNs and LSTMs, the topic-guided language model framework is agnostic to the class of neural language model used. We conclude by discussing how the results in this paper are relevant to researchers considering incorporating topic models into more powerful neural language models, such as transformers.

## 2 Study Design

Let $\mathbf{x}_{1:T} = \{x_1, \ldots, x_T\}$ be a sequence of tokens collectively known as a document, where each $x_t$ indexes one of $V$ words in a vocabulary (words outside the vocabulary are mapped to a special out-of-vocabulary token). Given a corpus of documents, the goal of language modeling is to learn a model $p(\mathbf{x}_{1:T})$ that approximates the probability of observing a document.

A document can be modeled autoregressively using the chain rule of probability,

$$p(\mathbf{x}_{1:T}) = \prod_{t=1}^{T} p(x_t \mid \mathbf{x}_{<t}), \tag{1}$$

where $\mathbf{x}_{<t}$ denotes all the words in a document before $t$. A language model parameterizes the predictive distribution of the next word, $p_{\boldsymbol{\mu}}(x_t \mid \mathbf{x}_{<t})$, with a set of parameters $\boldsymbol{\mu}$. Given a set of documents indexed by $D_{\text{train}}$, we compute a parameter estimate $\hat{\boldsymbol{\mu}}$ by maximizing the log likelihood objective,

$$\sum_{d=1}^{D_{\text{train}}} \sum_{t=1}^{T_d} \log p_{\boldsymbol{\mu}}(x_{d,t} \mid \mathbf{x}_{d,<t}),$$

with respect to $\boldsymbol{\mu}$. Language models are evaluated using perplexity on a held-out set of documents. With $D_{\text{test}}$ as the index set of the test documents, perplexity is defined as

$$\exp\left\{ -\frac{1}{\sum_d T_d} \sum_{d=1}^{D_{\text{test}}} \sum_{t=1}^{T_d} \log p_{\hat{\boldsymbol{\mu}}}(x_{d,t} \mid \mathbf{x}_{d,<t}) \right\}.$$

---

[1]https://github.com/carolinazheng/revisiting-tglms

Perplexity is the inverse geometric average of the likelihood of observing each word in the set of test documents under the fitted model; a lower perplexity indicates a better model fit.

## 3 Language Models and Topic Models

Here, we provide an overview of the two components of topic-guided language models: neural language models and topic models. The topic-guided language model literature has focused on models based on RNNs and LSTMs, which are the neural language models we focus on here.

**RNN language model.** A recurrent neural network (RNN) language model (Mikolov et al., 2010) defines each conditional probability in Equation (1) as

$$\mathbf{h}_{t-1} = f(x_{t-1}, \mathbf{h}_{t-2}) \tag{2}$$

$$p_{\mathrm{RNN}}(x_t \mid \mathbf{x}_{<t}) = \mathrm{softmax}(\mathbf{W}^\intercal \mathbf{h}_{t-1}), \tag{3}$$

where $\mathbf{W} \in \mathbb{R}^{D \times V}$ and $\mathbf{h}_t \in \mathbb{R}^D$. The hidden state $\mathbf{h}_{t-1}$ summarizes the information in the preceding sequence, while the function $f$ combines $\mathbf{h}_{t-1}$ with the word at time $t$ to produce a new hidden state, $\mathbf{h}_t$. The function $f$ is parameterized by a recurrent neural network (RNN).

The parameter $\mathbf{W}$ and the RNN model parameters are trained by maximizing the log likelihood of training documents using backpropagation through time (BPTT) (Williams & Peng). (In practice, the backpropagation of gradients is truncated after a specified sequence length.) The model directly computes the predictive distribution of the next word, $p(x_t \mid \mathbf{x}_{<t})$.

The baselines use the widely used RNN architecture, the LSTM (Hochreiter & Schmidhuber, 1997), as the language model, which we call LSTM-LM (Zaremba et al., 2014). The LSTM architecture is described in Appendix A.

To make full use of the document context, it is natural to condition on all previous words of the document when computing $p(x_t \mid \mathbf{x}_{<t})$. Even when the full document does not fit into memory, this can be done at no extra computational cost by storing the previous word's hidden state ($\mathbf{h}_{t-2}$ in Equation (2)) (Melis et al., 2017). This is our main baseline.

One can also define $\mathbf{x}_{<t}$ to be only the previous words in the current sentence. In this scenario, the model will not condition on all prior words in the document. This is the LSTM-LM baseline used in many TGLM papers (Lau et al., 2017; Guo et al., 2020; Rezaee & Ferraro, 2020). We call this model the sentence-level LSTM-LM.

**Topic model.** Another way to model a document is with a bag-of-words model that represents documents as word counts. One such model is a probabilistic topic model, which assumes the observed words are conditionally independent given a latent variable $\boldsymbol{\theta}$. In a topic model, the probability of a document is

$$p(\mathbf{x}_{1:T}) = \int \prod_{i=1}^T p(x_i \mid \boldsymbol{\theta}) p(\boldsymbol{\theta}) d\boldsymbol{\theta}, \tag{4}$$

where $p(\boldsymbol{\theta})$ is a prior distribution on $\boldsymbol{\theta}$ and $p(x \mid \boldsymbol{\theta})$ is the likelihood of word $x$ conditional on $\boldsymbol{\theta}$.

A widely used probabilistic topic model is Latent Dirichlet Allocation (LDA) (Blei et al., 2003). LDA posits that a corpus of text is comprised of $K$ latent topics. Each document $d$ contains a distribution over topics, $\boldsymbol{\theta}_d$, and each topic $k$ is associated with a distribution over words, $\boldsymbol{\beta}_k$. These two terms combine to form the distribution of each word in a document.

The generative model for LDA is:

1. Draw $K$ topics: $\boldsymbol{\beta}_1, \ldots, \boldsymbol{\beta}_K \sim \mathrm{Dirichlet}_V(\boldsymbol{\gamma})$.
2. For each document:
    (a) Draw topic proportions,
    $\boldsymbol{\theta} \sim \mathrm{Dirichlet}_K(\boldsymbol{\alpha})$.

(b) For each word $x_1, \ldots, x_T$:
    i. Draw topic indicator,
       $z_{x_t} \sim \text{Categorical}(\boldsymbol{\theta})$.
    ii. Draw word, $x_t \sim \text{Categorical}(\boldsymbol{\beta}_{z_{x_t}})$.

Since each word is drawn conditionally independent of the preceding words in the document, LDA is not able to capture word order or syntax. However, it can capture document-level patterns since the topic for each word is drawn from a document-specific distribution. Practitioners typically rely on approximate posterior inference to estimate the LDA posterior. The most common methods are Gibbs sampling (Griffiths & Steyvers, 2004) or variational inference (Blei et al., 2003).

After approximating the posterior distribution over topics from the training documents, the next-word posterior predictive distribution is

$$p_{\text{LDA}}(x_t \mid \mathbf{x}_{<t}) = \int p(x_t \mid \boldsymbol{\theta}) p(\boldsymbol{\theta} \mid \mathbf{x}_{<t}) d\boldsymbol{\theta}. \tag{5}$$

Given words $\mathbf{x}_{<t}$ from a document, one can use approximate posterior inference to estimate the topic proportions posterior, $p(\boldsymbol{\theta} \mid \mathbf{x}_{<t})$, and then draw Monte Carlo samples of $\boldsymbol{\theta}$ to estimate the predictive distribution.

## 4 Topic-Guided Language Model

We now discuss topic-guided language models (TGLMs), which are a class of language models that combine topic models and neural language models. TGLMs were initially proposed to combine the fluency of neural language models with the document modeling capabilities of topic models. Dieng et al. (2017) and Lau et al. (2017), who propose two of the models that we study here, argue that long-range dependency in language is captured well by topic models. Subsequent TGLM papers build on Dieng et al. (2017) and Lau et al. (2017), but differ from these previous works in evaluation setting (Wang et al., 2018; Rezaee & Ferraro, 2020; Guo et al., 2020).

Topic-guided language models can be divided into two frameworks, differing in whether they model the document's bag-of-words counts in addition to the typical next-word prediction objective. In this section, we discuss the two frameworks: a topic-biased language model and a joint topic and language model. The graphical structure of these models are shown in Figure 1.

### 4.1 Topic-Biased Language Models

A topic-biased language model defines the next-word probability to be the sum of two terms: a linear transformation of the hidden state, as in an RNN, and the distribution of words according to a document's topics, as in a topic model.

Each document follows the data generating mechanism below:

1. Draw topic vector, $\boldsymbol{\theta} \sim \text{Dirichlet}_K(\cdot)$.

2. For each word $x_1, \ldots, x_T$:
    (a) $\mathbf{h}_t = \text{RNN}(x_t, \mathbf{h}_{t-1})$.
    (b) Draw $\ell_t \sim \text{Bernoulli}(\sigma(\mathbf{u}^\intercal \mathbf{h}_t))$.
    (c) Draw $z_t \sim \text{Categorical}(\boldsymbol{\theta})$.
    (d) Draw $x_{t+1} \propto \exp(\mathbf{W}^\intercal \mathbf{h}_t + (1 - \ell_t)\boldsymbol{\beta}_{z_t})$.

Here, $\sigma(\cdot)$ denotes the sigmoid function. The model parameters are the parameters of the RNN, the weights $\mathbf{W} \in \mathbb{R}^{D \times V}$ and $\mathbf{u} \in \mathbb{R}^D$, and the topics $\boldsymbol{\beta}_1, \ldots, \boldsymbol{\beta}_K \in \mathbb{R}^V$. The latent variable $\boldsymbol{\theta}$ determines the document's topic proportions.

Of the two additive terms in a word's likelihood, the RNN term encourages fluency and syntax while the topic modeling term can be understood as a bias toward the document's global structure. Since topic models

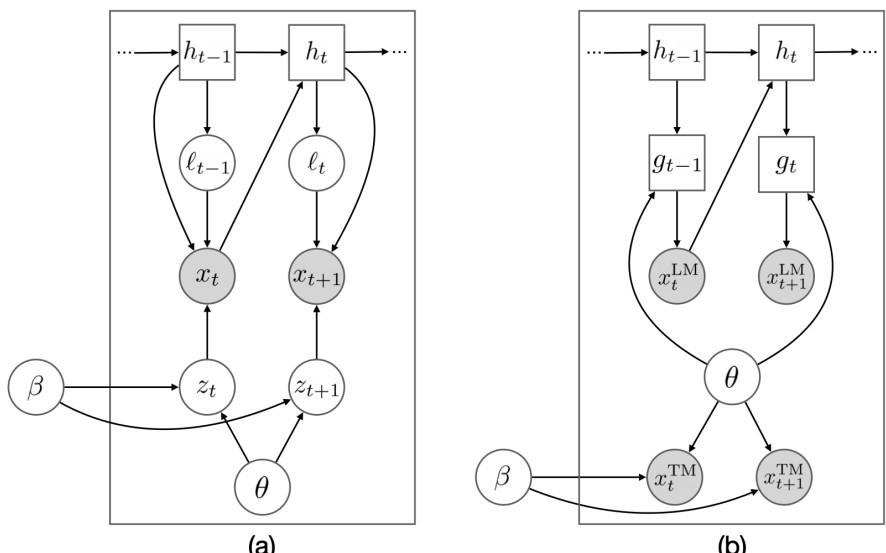

Figure 1: Graphical model representations of the two frameworks of topic-guided language models. (a) is the topic-biased language model. (b) is the joint topic and language model. Circles denote random variables, while squares denote deterministic variables. Shading indicates that the variable is observed.

struggle with modeling very common words ("stop words") (Wallach et al., 2009), a word's likelihood only includes the topic modeling term if it is not predicted to be a stop word ($\ell_t = 0$). The realizations of the stop word indicators are observed during training ($\ell_t = 1$ if $x_{t+1}$ belongs to a list of stop words, and 0 otherwise). During prediction, the stop word indicators are treated as latent variables and are marginalized out. Hence, the topic-biased language models learn to interpolate between a standard language model's predictions and topics.

**TopicRNN.** TopicRNN (Dieng et al., 2017) approximates Step 2(d) by marginalizing $z_t$ before normalizing:

$$p_{\text{TRNN}}(x_{t+1} \mid h_t, \boldsymbol{\theta}) \propto \exp(\mathbb{E}[\mathbf{W}^\mathsf{T}\mathbf{h}_t + (1 - \ell_t)\boldsymbol{\beta}_{z_t}])$$
$$= \exp(\mathbf{W}^\mathsf{T}\mathbf{h}_t + (1 - \ell_t)\boldsymbol{\beta}^\mathsf{T}\boldsymbol{\theta}).$$

The topic matrix $\boldsymbol{\beta} \in \mathbb{R}^{K \times V}$ contains the topic vectors $\boldsymbol{\beta}_1, \ldots, \boldsymbol{\beta}_K$ as rows. Additionally, in Step 1, TopicRNN draws $\boldsymbol{\theta}$ from a standard Gaussian, rather than a Dirichlet distribution.

**VRTM.** VRTM (Rezaee & Ferraro, 2020) (short for Variational Recurrent Topic Model) exactly computes Step 2(d) by marginalizing $z_t$ after normalizing:

$$p_{\text{VRTM}}(x_{t+1} \mid \mathbf{h}_t, \boldsymbol{\theta}) = \mathbb{E}[\text{softmax}(\mathbf{W}^\mathsf{T}\mathbf{h}_t + (1 - \ell_t)\boldsymbol{\beta}_{z_t})]$$
$$= \sum_{k=1}^{K} \boldsymbol{\theta}_k * \text{softmax}(\mathbf{W}^\mathsf{T}\mathbf{h}_t + (1 - \ell_t)\boldsymbol{\beta}_k).$$

This makes VRTM a mixture-of-RNNs (Yang et al., 2017), where the mixture proportions are determined by $\boldsymbol{\theta}$.

**Inference.** The model parameters for topic-biased language models are learned using variational inference (Wainwright et al., 2008; Blei et al., 2017). We provide a high-level overview of the method here.

The goal of variational inference is to approximate the posterior of the latent variable $\boldsymbol{\theta}$, $p(\boldsymbol{\theta} \mid \mathbf{x}_{1:T})$, with a learned distribution $q_\phi(\boldsymbol{\theta})$, called the variational distribution. To fit $q_\phi(\boldsymbol{\theta})$, variational inference minimizes

the KL divergence between the two distributions. This is equivalent to maximizing a lower bound of the marginal log likelihood, or evidence lower bound (ELBO):

$$\log p(\mathbf{x}_{1:T}) = \log \int p_{\mu}(\mathbf{x}_{1:T} \mid \boldsymbol{\theta}) p(\boldsymbol{\theta}) d\boldsymbol{\theta}$$
$$\geq \mathbb{E}_{q_{\phi}(\boldsymbol{\theta})}[\log p_{\boldsymbol{\mu}}(\mathbf{x}_{1:T} \mid \boldsymbol{\theta})] - \mathrm{KL}(q_{\phi}(\boldsymbol{\theta}) \| p(\boldsymbol{\theta})).$$

The ELBO contains two terms: a reconstruction loss, which is the expected log probability of the data under $q_{\phi}(\boldsymbol{\theta})$, and the KL divergence between the variational distribution and the prior on $\boldsymbol{\theta}$. By maximizing the ELBO, we can simultaneously learn the model parameters and the variational distribution parameters, represented by $\boldsymbol{\mu}$ and $\boldsymbol{\phi}$ respectively. In order to share the learned variational parameters, the variational distribution is defined to be a function of the data, i.e., we learn $q_{\phi}(\boldsymbol{\theta} \mid \mathbf{x}_{1:T})$, where $q_{\phi}$ is parameterized by a neural network.

In practice, the ELBO is maximized with respect to parameters $\boldsymbol{\mu}$ and $\boldsymbol{\phi}$ using backpropagation. The expectation is estimated using samples from $q_{\phi}(\boldsymbol{\theta} \mid \mathbf{x}_{1:T})$ and the KL can often be computed analytically (e.g., when both distributions are Gaussians) (Kingma & Welling, 2014).

**Prediction.** For both models, the next-word predictive distribution is

$$p(x_t \mid \mathbf{x}_{<t}) = \mathbb{E}_{p(\boldsymbol{\theta} \mid \mathbf{x}_{<t})}[p(x_t \mid \mathbf{x}_{<t}, \boldsymbol{\theta})]. \tag{6}$$

Using a learned variational distribution in place of the exact posterior, we approximate the expectation using its mean, i.e., let $\hat{\boldsymbol{\theta}} = \mathbb{E}[q_{\phi}(\boldsymbol{\theta} \mid \mathbf{x}_{<t})]$. Then $p(x_t \mid \mathbf{x}_{<t}) \approx p(x_t \mid \mathbf{x}_{<t}, \hat{\boldsymbol{\theta}})$. For computation reasons, in practice, $\hat{\boldsymbol{\theta}}$ is only updated in a sliding window (i.e., every $N$ words).

## 4.2 Joint Topic and Language Model

A joint topic and language model learns the topic model and language model simultaneously, essentially fitting two views of the same data. The two models share the document-level latent variable $\boldsymbol{\theta}$.

Each document during training has a pair of representations, its bag-of-words $\mathbf{x}_{1:T}^{\mathrm{TM}}$ and its word sequence $\mathbf{x}_{1:T}^{\mathrm{LM}}$, generated by the topic model and the language model, respectively. For each document, a basic version of the data generating mechanism is:

1. Draw topic vector, $\boldsymbol{\theta} \sim \mathrm{Dirichlet}_K(\cdot)$.
2. Draw the bag-of-words $\mathbf{x}_{1:T}^{\mathrm{TM}}$ from a topic model (Section 3):
   (a) $\mathbf{x}_{1:T}^{\mathrm{TM}} \sim \mathrm{TopicModel}(\boldsymbol{\theta})$.
3. For each word $x_1^{\mathrm{LM}}, \ldots, x_T^{\mathrm{LM}}$:
   (a) $\mathbf{h}_t = \mathrm{RNN}(x_t^{\mathrm{LM}}, \mathbf{h}_{t-1})$.
   (b) $\mathbf{g}_t = a(\mathbf{h}_t, \boldsymbol{\theta})$.
   (c) Draw $x_{t+1}^{\mathrm{LM}} \propto \exp(\mathbf{W}^{\mathsf{T}} \mathbf{g}_t)$.

Here, the latent variable $\boldsymbol{\theta}$ determines the document's topic proportions in the topic model. In the language model, the hidden state $\mathbf{h}_t$ is combined with $\boldsymbol{\theta}$ in a differentiable function $a$, usually the Gated Recurrent Unit (Cho et al., 2014). The GRU architecture is described in Appendix B.

The model parameters are the parameters of the topic model, the parameters of the RNN, the parameters of $a$, and the weights $\mathbf{W} \in \mathbb{R}^{D \times V}$.

**TDLM.** TDLM (Lau et al., 2017) (short for Topically Driven Language Model) is a variant of the model outlined above. There are two major differences. First, $\boldsymbol{\theta}$ is not considered to be a latent variable. Instead, an encoder function maps a bag-of-words to $\boldsymbol{\theta}$. In the topic model, the bag-of-words used is from the entire document, i.e., $\boldsymbol{\theta}^{\mathrm{TM}} = \mathrm{enc}(\mathbf{x}_{1:T}^{\mathrm{TM}})$.

Second, the language model component of the data generating process (Step 3) uses a different $\boldsymbol{\theta}$ than the topic modeling component, which we call $\boldsymbol{\theta}^{\mathrm{LM}}$. To prevent the model from memorizing the current sentence,

$\boldsymbol{\theta}^{\mathrm{LM}}$ is computed from the document bag-of-words excluding the current sentence. In the language model, if $j$ is the index set of the words in the current sentence, $\boldsymbol{\theta}^{\mathrm{LM}} = \mathrm{enc}(\mathbf{x}_{1:T\backslash j}^{\mathrm{TM}})$.

**Inference.** TDLM is trained by maximizing the log likelihood of the topic model and the language model jointly. The objective, $\mathcal{L}_{\mathrm{TDLM}}$, is:

$$\mathcal{L}_{\mathrm{TM}} = \log p(\mathbf{x}_{1:T}^{\mathrm{TM}} \mid \boldsymbol{\theta}^{\mathrm{TM}})$$
$$\mathcal{L}_{\mathrm{LM}} = \sum_t \log p(x_t^{\mathrm{LM}} | \mathbf{x}_{<t}^{\mathrm{LM}}, \boldsymbol{\theta}^{\mathrm{LM}})$$
$$\mathcal{L}_{\mathrm{TDLM}} = \mathcal{L}_{\mathrm{TM}} + \mathcal{L}_{\mathrm{LM}}.$$

Although the original model only conditions on previous words in the current sentence when forming $\mathcal{L}_{\mathrm{LM}}$, we condition on all prior words in the document because it improves performance.

**Prediction.** Let $\hat{\boldsymbol{\theta}} = \mathrm{enc}(\mathbf{x}_{<t}^{\mathrm{TM}})$. The next-word predictive distribution is

$$p(x_t^{\mathrm{LM}} \mid \mathbf{x}_{<t}^{\mathrm{LM}}) = p(x_t^{\mathrm{LM}} \mid \mathbf{x}_{<t}^{\mathrm{LM}}, \hat{\boldsymbol{\theta}}), \tag{7}$$

which is defined in Step 3 of the data generating process. In practice, like prediction for the topic-biased LMs, we recompute $\hat{\boldsymbol{\theta}}$ in a sliding window.

**rGBN-RNN.** rGBN-RNN (Guo et al., 2020) is an extension of the model outlined in this section. In rGBN-RNN's topic model, each sentence $j$ has a unique bag-of-words: it is the document's bag-of-words with the sentence excluded, denoted $\mathbf{x}_{1:T\backslash j}^{\mathrm{TM}}$. In Step 1 of the data generating mechanism, a different topic vector is drawn sequentially for each sentence. For sentences $j = 1, \ldots, J$, where $J$ is the total number of sentences,

$$\boldsymbol{\theta}_j \sim \mathrm{Gamma}(\boldsymbol{\Pi}\boldsymbol{\theta}_{j-1}, \tau_0),$$

where $\boldsymbol{\Pi}$ and $\tau_0$ are model parameters. In Step 2, for each sentence $j$, its bag-of-words is drawn:

$$\mathbf{x}_{1:T\backslash j}^{\mathrm{TM}} \sim \mathrm{Poisson}(\boldsymbol{\Phi}\boldsymbol{\theta}_j),$$

where $\boldsymbol{\Phi}$ is a model parameter.

For the language modeling component (Step 3 of the data generating mechanism), rGBN-RNN generates individual sentences. In Step 3, each sentence $j$ is conditionally independent of the other sentences, given its corresponding topic vector, $\boldsymbol{\theta}_j$. In other words,

$$p(x_{j_t}^{\mathrm{LM}} | \mathbf{x}_{<j_t}^{\mathrm{LM}}, \boldsymbol{\theta}_j) = p(x_{j_t}^{\mathrm{LM}} | \mathbf{x}_{j,<t}^{\mathrm{LM}}, \boldsymbol{\theta}_j), \tag{8}$$

where $x_{j_t}^{\mathrm{TM}}$ is the $t$'th word of sentence $j$, $\mathbf{x}_{<j_t}^{\mathrm{LM}}$ denotes all the words in document before the $t$'th word of the $j$'th sentence, and $\mathbf{x}_{j,<t}^{\mathrm{LM}}$ denotes only the words in the $j$'th sentence before the $t$'th word.

rGBN-RNN also introduces multiple stochastic layers to both the topic model and language model, which is simplified to one layer in this exposition. In the experiments, we use the full original model.

**Inference.** rGBN-RNN is trained using a combination of variational inference and stochastic gradient MCMC (Guo et al., 2018). We refer the reader to Guo et al. (2020) for further mathematical details of the model and inference algorithm.

**Prediction.** For each sentence $j$, the next-word predictive distribution is

$$p(x_{j,t}^{\mathrm{LM}} \mid \mathbf{x}_{<j_t}^{\mathrm{LM}}) = \mathbb{E}_{p(\boldsymbol{\theta}_j \mid \mathbf{x}_{<j_1}^{\mathrm{TM}})}[p(x_{j,t}^{\mathrm{LM}} \mid \mathbf{x}_{j,<t}^{\mathrm{LM}}, \boldsymbol{\theta}_j)].$$

The expectation is approximated using a sample from the approximate posterior of $\boldsymbol{\theta}_j$ computed during inference. The topic model parameters, $\boldsymbol{\Phi}$ and $\boldsymbol{\Pi}$, are similarly marginalized out via MCMC sampling (see Guo et al. (2020) for more details).

Table 1: The LSTM-LM baseline matches or exceeds topic-guided language models in held-out perplexity. In parentheses are standard deviations estimated by averaging three runs with different random seeds. The comparable baseline to each topic-guided language model is the LSTM-LM (Zaremba et al., 2014) above it in the table. Parameter counts are listed in Appendix E.

| Model | LSTM Size | Topic Size | Perplexity | | | |
|---|---|---|---|---|---|---|
| | | | APNEWS | IMDB | BNC | WT-2 |
| LSTM-LM (sentence-level) | 600 | – | 65.0 (0.3) | 76.8 (0.1) | 112.9 (0.3) | 115.3 (0.8) |
| LSTM-LM | 600 | – | 56.5 (0.3) | 73.1 (0.2) | 96.4 (0.1) | 90.7 (0.7) |
| TopicRNN (Dieng et al., 2017) | 600 | 100 | 56.6 (0.3) | 73.0 (0.1) | 96.8 (0.2) | 93.2 (0.6) |
| VRTM (Rezaee & Ferraro, 2020) | 600 | 50 | 56.8 (0.3) | 73.6 (0.2) | 96.3 (0.2) | 90.8 (0.6) |
| LSTM-LM | 600 (+GRU) | – | 53.5 (0.2) | 68.8 (1.3) | 91.2 (0.1) | 89.9 (0.8) |
| TDLM (Lau et al., 2017) | 600 (+GRU) | 100 | 53.7 (0.1) | 68.8 (0.1) | 91.4 (0.2) | 90.4 (0.7) |
| LSTM-LM | 600x3 | – | 51.9 (0.4) | 66.6 (0.8) | 88.8 (0.3) | 89.5 (0.7) |
| rGBN-RNN (Guo et al., 2020) | 600x3 | 100-80-50 | 52.6 (0.3) | 64.8 (0.2) | 97.7 (1.1) | – |

## 5 Experiments

In this section, we detail the reproducibility study and results. We also investigate the quality of learned topics and probe the LSTM-LM's hidden representations to find the amount of retained topic information.

### 5.1 Reproducibility Study

We evaluate the held-out perplexity of four TGLMs and corresponding LSTM-LM baselines on four document-level corpora.

**Datasets.** We use four publicly available natural language datasets: APNEWS,[2] IMDB (Maas et al., 2011), BNC (Consortium, 2007), and WikiText-2 (Merity et al., 2017). We follow the training, validation, and test splits from Lau et al. (2017) and Merity et al. (2017). Details about the datasets and preprocessing steps are in Appendix C.

**Models.** The LSTM-LM is described in Section 2. The four topic-guided language models are TDLM (Lau et al., 2017), TopicRNN (Dieng et al., 2017), rGBN-RNN (Guo et al., 2020), and VRTM (Rezaee & Ferraro, 2020), and are described in Section 4.

We implement TopicRNN (Dieng et al., 2017), TDLM (Lau et al., 2017), and VRTM (Rezaee & Ferraro, 2020) from scratch. For rGBN-RNN (Guo et al., 2020), we use the publicly available codebase and make minimal adjustments to the codebase to ensure that preprocessing and evaluation are consistent. Some other topic-guided language models do not have public code (Wang et al., 2018; Tang et al., 2019; Wang et al., 2019) and are not straightforward to implement. These models are not part of the study, but their architecture is similar to that of Lau et al. (2017), which we compare to.

For all LSTM-LM baselines, we use a hidden size of 600, word embeddings of size 300 initialized with Google News word2vec embeddings (Mikolov et al., 2013), and dropout of 0.4 between the LSTM input and output layers (and between the hidden layers for the 3-layer models). For the four TGLMs we study, we use the same settings as LSTM-LM for the LSTM components. For the additional TGLM-specific components, we use the architectures and settings from the original papers, except for small details reported in Appendix D to make certain settings consistent across models.[3]

To obtain comparable baselines to all TGLMs studied, we train three LSTM-LMs of varying sizes. The default baseline is a 1-layer LSTM-LM. To control for the additional GRU layer in the language model

---

[2]https://www.ap.org/en/
[3]The one additional change is that Guo et al. (2020) reports a 600-512-256 size model, but their public code only supports 3-layer models with same-size RNN layers, so we use 600-600-600.

component of TDLM, we train a 1-layer LSTM-LM with a GRU layer between the LSTM output and the output embedding layer. To compare to rGBN-RNN, a hierarchical model, we train a 3-layer LSTM-LM. Finally, we also compare to a baseline considered in prior work, an LSTM-LM conditioned only on previous words in the same sentence. We call this model the sentence-level LSTM-LM.

**Training Details.** We train the RNN components using truncated backpropagation through time with a sequence length of 30. For sequences within the same document (or sentence for the sentence-level LSTM-LM), if $\mathbf{h}_i$ is the final hidden state computed by the RNN for the $i^{\text{th}}$ sequence, we initialize the initial hidden state for the $(i + 1)^{\text{th}}$ sequence with stop_gradient($\mathbf{h}_i$). Although the TDLM and rGBN-RNN models assume conditional independence between sentences, we found that in practice, keeping hidden states between sentences improved their performance.

Following Lau et al. (2017), Rezaee & Ferraro (2020), and Guo et al. (2020), we use the Adam optimizer with a learning rate of 0.001 on APNEWS, IMDB, and BNC. For WikiText-2, we follow Merity et al. (2017) and use stochastic gradient descent; the initial learning rate is 20 and is divided by 4 when validation perplexity is worse than the previous iteration. The models are trained until validation perplexity does not improve for 5 epochs and we use the best validation checkpoint. The models in our codebase train to convergence within three days on a single Tesla V100 GPU. rGBN-RNN, trained using its public codebase, trains to convergence within one week on the same GPU. We do not include WikiText-2 results for rGBN-RNN because its perplexity did not decrease during training.

**Results.** Table 1 shows the results. After controlling for language model size, *the LSTM-LM baseline consistently matches or outperforms topic-guided language models with the same number of parameters.* Although most topic-guided language models improve over a baseline considered in prior work — the sentence level LSTM-LM — they are matched by an LSTM which, like topic-guided language models, conditions on all prior words in a document. As discussed in Section 2, this is a standard practice that can be performed at no extra computational cost.

## 5.2 Probing the RNN hidden states

The motivation for topic-guided language models is to augment language models with document-level information captured by topic models (Dieng et al., 2017). To assess whether topic models are adding useful information, we perform a probe experiment. In NLP, probe tasks are designed to understand the extent to which linguistic structure is encoded in the representations produced by black-box models (Alain & Bengio, 2016; Conneau et al., 2018; Hewitt & Liang, 2019; Pimentel et al., 2020). In this case, we probe the baseline LSTM-LM's hidden representations to assess how much topic information it already captures.

Specifically, we evaluate whether an LSTM-LM's hidden representation of the document's first $t$ words, $\mathbf{h}_t$, is predictive of the topic vector estimated from the document's first $t$ words, $\boldsymbol{\theta}_t$, of a topic-guided language model. We evaluate TDLM since it is the best performing topic-guided language model and has the highest quality topics (see Section 5.3). We also evaluate TopicRNN, a topic-guided language model with lower quality topics.

We use the fitted models from Section 5.1 to create training data for the probe experiment. The input is the LSTM-LM's initial hidden state $\mathbf{h}_t$ for each sequence in a document (in this experiment, we define a sequence to be a 30-word chunk). The output is TDLM's topic proportions at the sequence, transformed with inverse-softmax to ensure it is real-valued: $\tilde{\boldsymbol{\theta}}_t = \log(\boldsymbol{\theta}_t) - \sum_{j=1}^{K} \log(\boldsymbol{\theta}_{t,j})$, where $j$ indexes the dimension of $\boldsymbol{\theta}$.

In the experiment, a linear model is trained to predict $\tilde{\boldsymbol{\theta}}_t$ from $\mathbf{h}_t$. The loss function is mean squared error summed across the topic components. The held-out prediction quality of this model (or "probe") can be viewed as a proxy for mutual information between the LSTM-LM's hidden state and the topic proportion vector $\boldsymbol{\theta}$, which is not easily estimable. For each topic-guided language model, we also run a baseline experiment where we probe a randomly initialized LSTM-LM that was not fit to data.

Table 2: The probe experiment reveals that the hidden state of an LSTM-LM is predictive of the topic information of two topic-guided language models, TDLM and TopicRNN, on held-out data. We also train baselines where the probe data is from a randomly initialized LSTM-LM. Standard deviations are listed in Appendix F.

| Target | Data | APNEWS | | | IMDB | | | BNC | | | WT-2 | | |
|---|---|---|---|---|---|---|---|---|---|---|---|---|---|
| | | Acc-1 | Acc-5 | $R^2$ | Acc-1 | Acc-5 | $R^2$ | Acc-1 | Acc-5 | $R^2$ | Acc-1 | Acc-5 | $R^2$ |
| TDLM | init. | .036 | .135 | .134 | .028 | .132 | .023 | .098 | .294 | .073 | .128 | .379 | .018 |
| TDLM | trained | .340 | .681 | .621 | .180 | .449 | .400 | .314 | .638 | .379 | .510 | .858 | .352 |
| TopicRNN | init. | .075 | .224 | .022 | .086 | .259 | .014 | .077 | .195 | .017 | .197 | .442 | -.010 |
| TopicRNN | trained | .210 | .540 | .232 | .238 | .575 | .231 | .172 | .424 | .170 | .208 | .486 | .067 |

Table 2 shows the results of the probe experiment. The linear model can reconstruct TDLM's topic proportion vector to some extent; 62% of the variance in a held-out set of APNEWS topic proportions can be explained by the LSTM's hidden state. Moreover, the hidden state predicts TDLM's largest topic for between 18% to 51% of test sequences across datasets, and improves 15% to 30% over the initialization-only baseline. These accuracies indicate that the LSTM-LM has learned to capture TDLM's most prevalent topic. Compared to TDLM, the TopicRNN probe exhibits a smaller improvement over its baseline. Nevertheless, the probe task shows that a notable amount of broader topic information is captured by the baseline LSTM's hidden state.

## 5.3 Topic quality

Although the predictions from topic-guided language models are matched or exceeded by an LSTM-LM baseline, it is possible that the learned topics will still be useful to practitioners. We compare the topics learned by topic-guided language models to those learned by a classical topic model, LDA (Blei et al., 2003). While LDA's next word predictions are worse than those of neural topic models (Dieng et al., 2020), its topics may be more interpretable. To assess the quality of learned topics, we compute an automated metric that correlates with human judgements of topic coherence (Aletras & Stevenson, 2013; Lau et al., 2014).

Automated coherence can be estimated using normalized pointwise mutual information (NPMI) scores. The NPMI score of a topic from its $N$ top words is defined as

$$\binom{N}{2}^{-1} \sum_{i=1}^{N-1} \sum_{j=i+1}^{N} \frac{\log \frac{p(w_i, w_j)}{p(w_i)p(w_j)}}{-\log p(w_i, w_j)}, \tag{9}$$

and ranges from $-1$ to $1$. To compute the word co-occurrence statistics (estimates of $p(w_i)$ and $p(w_i, w_j)$ in Equation (9)) we use the corresponding dataset as the reference corpus. To obtain the model-level coherence, we average the scores from the top 5/10/15/20 topic words for each topic, then average over all topics.[4]

The coherence for each model is in Table 3. The topic-biased language models (TopicRNN and VRTM) learn largely incoherent topics, while only TDLM (Lau et al., 2017) achieves comparable coherences to LDA in two out of the four corpora, APNEWS and BNC. Since the quality of automated topic evaluation metrics has been disputed for neural topic models (Hoyle et al., 2021), Appendix G includes the top words from randomly sampled topics. The joint topic and language models (TDLM and rGBN-RNN) learn qualitatively acceptable topics, while VRTM fails to learn distinct topics entirely.

## 6 Discussion

This reproducibility study compares topic-guided language models to LSTM baselines. We find that the baselines outperform the topic-guided language models in predictive performance. This finding differs from

---

[4]We use gensim (Rehurek & Sojka, 2011) to calculate NPMI scores, with a window size of 10. In processing the reference corpora, we retain only terms that exist in the topic model vocabulary.

Table 3: Topic coherences across corpora and models. The topic-guided language models do not learn topics that are more coherent than LDA. In parentheses are standard deviations in coherence from training with three different random seeds per model. Note we do not include VRTM in the table because the model did not learn distinct topics (see Table 7).

| | | Coherence | | |
|---|---|---|---|---|
| Model | APNEWS | IMDB | BNC | WT-2 |
| LDA | .125 (.00) | .080 (.01) | .124 (.00) | .093 (.01) |
| TDLM | .176 (.00) | .011 (.02) | .104 (.01) | -.026 (.03) |
| TopicRNN | -.330 (.00) | -.327 (.01) | -.293 (.01) | -.311 (.00) |
| rGBN-RNN | .047 (.01) | .002 (.01) | -.017 (.01) | – |

the results reported in the topic-guided language modeling literature, which shows improvements over baselines. In general, these differences are due to weaker baselines in the literature and a form of evaluation that considers future words.

**Baselines.** The baseline compared to in most prior work (Lau et al., 2017; Rezaee & Ferraro, 2020; Guo et al., 2020) is the sentence-level LSTM-LM, which we report as the weakest model in Table 1; this baseline does not condition on all words in a document's history. Similarly, the baseline in Dieng et al. (2017) does not condition on the history beyond a fixed-context window. In contrast, the LSTM-LM baseline in this work conditions on all previous words in the document during training and evaluation. Our findings suggest that the predictive advantage of topic-guided language models stems from conditioning on all prior words in a document via a representation of topic proportions.

Additionally, topic-guided language models typically augment their language model component with additional parameters. We only compare topic-guided language models to baselines with a similar number of language model parameters. TDLM (Lau et al., 2017) adds parameters to its language model via an additional GRU; however, TDLM was originally compared to a baseline without this module. We find that TDLM's predictive advantage fades when scaling the LSTM-LM baseline to match TDLM's language model size.

**Evaluation.** Different papers have evaluated the performance of topic-guided language models in different ways. In this paper, we standardize the evaluation of models and their baselines to make sure results are comparable.

As described in Section 4, topic-guided language models use a representation of the full document to estimate the topic proportions vector $\theta$ during training. During evaluation, $\theta$ must be estimated using only previous document words. Otherwise, a model would be looking ahead when making next-word predictions.[5]

Some methods in the TGLM literature have conditioned on future words in their evaluation, making them incomparable to language models that only condition on previous words. For example, TDLM (Lau et al., 2017) is proposed as a sentence-level model, and thus the paper reports results when conditioning on future words. Additionally, VRTM (Rezaee & Ferraro, 2020) and rGBN-RNN (Guo et al., 2020) are proposed as document-level models, but in the respective evaluation scripts of their public codebases, $\theta$ is estimated using future words.

Finally, conditioning during evaluation isn't the only difference among these models. Some prior papers do not use consistent language model vocabulary sizes, which makes reported numbers incomparable. For example, VRTM employs a smaller vocabulary size than the baselines and other models it compares to. These discrepancies in evaluation may account for differences in reported results.

---

[5]We also ran experiments that corrected this mismatch by estimating $\theta$ using only previous document words in both training and evaluation, but found that this did not help performance.

## 7  Conclusion

We find that compared to a standard LSTM language model, topic-guided language models do not improve language modeling performance. The probe experiment shows that this is due to both the standard LSTM already possessing some level of topic understanding but also that capturing the exact topic vector is unnecessary for the task. For the two topic-biased language models, the quality of the learned topics is poor. This may be due to the choice of latent variable for the topic proportions vector $\boldsymbol{\theta}$, as previous work has found that using a Gaussian leads to low topic coherence, while using a Dirichlet with reparameterization gradients is prone to posterior collapse (Srivastava & Sutton, 2017; Burkhardt & Kramer, 2019).

While this study shows that current topic-guided language models do not improve next-word predictive performance, it is possible that incorporating a topic model can provide greater control or diversity in language model generations. Topic-guided language models can generate text conditional on topics (Lau et al., 2017; Guo et al., 2020); one potential direction for future work is a systematic investigation of controllability in topic-guided language models.

The topic-guided language modeling literature has focused on LSTMs, but we note that this framework is agnostic to the class of neural language model used. This means the same framework can be used to incorporate topic models into more powerful neural language models, such as transformers (Vaswani et al., 2017). However, if incorporating topic models into transformers does not improve predictive performance or provide meaningful latent variables, it is not necessarily because of architectural differences between transformers and LSTMs. Rather, the probing results in this paper indicate that neural language models are sufficiently expressive such that they already retain topic information. Transformers, which are more expressive than LSTMs, are likely even more capable of capturing topic information without explicitly modeling topics. Novel approaches are needed to enable joint learning of expressive neural language models and interpretable, topic-based latent variables.

### Acknowledgements

We thank Adji Dieng and the reviewers for their thoughtful comments and suggestions, which have greatly improved the paper. This work is supported by NSF grant IIS 2127869, ONR grants N00014-17-1-2131 and N00014-15-1-2209, the Simons Foundation, and Open Philanthropy.

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

Table 4: Dataset statistics.

| Dataset | Vocab Size | Training | | Validation | | Test | |
|---|---|---|---|---|---|---|---|
| | | Docs | Tokens | Docs | Tokens | Docs | Tokens |
| APNEWS | 34231 | 50K | 15M | 2K | 0.6M | 2K | 0.6M |
| IMDB | 36009 | 75K | 20M | 12.5K | 0.3M | 12.5K | 0.3M |
| BNC | 43703 | 15K | 18M | 1K | 1M | 1K | 1M |
| WikiText-2 | 33280 | 6182 | 2M | 620 | 218K | 704 | 246K |

Zhilin Yang, Zihang Dai, Ruslan Salakhutdinov, and William W. Cohen. Breaking the softmax bottleneck: A high-rank RNN language model. *CoRR*, 2017.

Wojciech Zaremba, Ilya Sutskever, and Oriol Vinyals. Recurrent neural network regularization, 2014.

## A  LSTM

The LSTM (Hochreiter & Schmidhuber, 1997) components are

$$i_t = \sigma(\mathbf{W}_i \mathbf{v}_t + \mathbf{U}_i \mathbf{h}_{t-1} + b_i)$$
$$f_t = \sigma(\mathbf{W}_f \mathbf{v}_t + \mathbf{U}_f \mathbf{h}_{t-1} + b_f)$$
$$o_t = \sigma(\mathbf{W}_o \mathbf{v}_t + \mathbf{U}_o \mathbf{h}_{t-1} + b_i)$$
$$\hat{\mathbf{c}}_t = \tanh(\mathbf{W}_c \mathbf{v}_t + \mathbf{U}_c \mathbf{h}_{t-1} + b_c)$$
$$\mathbf{c}_t = f_t \odot \mathbf{c}_{t-1} + i_t \odot \hat{\mathbf{c}}_t$$
$$\mathbf{h}_t = o_t \odot \tanh(\mathbf{c}_t).$$

The symbol $\odot$ denotes element-wise product, while $i_t, f_t, o_t$ are the input, forget, and output activations at time $t$. Additionally, $\mathbf{v}_t, \mathbf{h}_t, \mathbf{c}_t$ are the input word embedding, hidden state, and cell state at time $t$. Finally, $\mathbf{W}, \mathbf{U}, b$ are model parameters.

## B  GRU

The GRU (Cho et al., 2014) components are

$$z_t = \sigma(\mathbf{W}_z \mathbf{v}_t + \mathbf{U}_z \mathbf{h}_t + b_z)$$
$$r_t = \sigma(\mathbf{W}_r \mathbf{v}_t + \mathbf{U}_r \mathbf{h}_t + b_r)$$
$$\hat{\mathbf{h}}_t = \tanh(\mathbf{W}_h \mathbf{s} + \mathbf{U}_h(r_t \odot \mathbf{h}_t) + b_h)$$
$$\mathbf{h}'_t = (1 - z_t) \odot \mathbf{h}_t + z_t \odot \hat{\mathbf{h}}_t.$$

Here, $z_t$ and $r_t$ are the update and reset gate activations at time $t$. Meanwhile, $\mathbf{v}_t$ and $\mathbf{h}_t$ are the input vector and the hidden state at time $t$, while $\mathbf{W}, \mathbf{U}$, and $b$ are model parameters.

## C  Datasets

We evaluate on four publicly available corpora. APNEWS is a collection of Associated Press news articles from 2009 to 2016. IMDB is a set of movie reviews collected by Maas et al. (2011). BNC is the written portion of the British National Corpus (Consortium, 2007), which contains excerpts from journals, books, letters, essays, memoranda, news, and other types of text. WikiText-2 is a subset of the verified Good or Featured Wikipedia articles (Merity et al., 2017). A random subset of APNEWS and BNC is selected for the experiments.

Table 4 shows the dataset statistics. The data is preprocessed as follows. For WikiText-2, we use the standard vocabulary, tokenization, and splits from Merity et al. (2017). We determine documents based on section header lines in the data. The EOS token is prepended to the start of each document and added to the end of each document.

For APNEWS, IMDB, and BNC, documents are lowercased and tokenized using Stanford CoreNLP (Klein & Manning, 2003). Tokens that occur less than 10 times are replaced with the UNK token. The SOS token is prepended to the start of each document; the EOS token is appended to the end of each document. While we use the same vocabulary and tokenization as Lau et al. (2017), we do not add extra SOS and EOS tokens to the beginning and end of each sentence, so our perplexity numbers are not directly comparable to Lau et al. (2017); Rezaee & Ferraro (2020); Guo et al. (2020), who evaluate on the same datasets. When we redo the reproducibility experiment to use their original preprocessing settings, the trends in model performance are nearly identical to the results in this paper. We use the same splits as Lau et al. (2017).

Each model uses the same vocabulary for next-word prediction, so predictive performance is comparable across models. Models have different specifications for the number of words in the topic model component. For rGBN-RNN and TDLM, we follow the vocabulary preprocessing steps outlined in the respective papers. We exclude the top 0.1% most frequent tokens and tokens that appear in the Mallet stop word list. For TopicRNN and VRTM, we additionally exclude words that appear in less than 100 documents, following Wang et al. (2018).

## D   Experiment Settings

We train all models on single GPUs with a language model batch size of 64. The experiments can be replicated on an AWS Tesla V100 GPU with 16GB GPU memory. LSTM-LM, TopicRNN, VRTM, and TDLM are implemented in our codebase in Pytorch 1.12. We use the original implementation of rGBN-RNN, which uses Tensorflow 1.9.

We note differences between our experiment settings and the original papers here. All other settings are the same as in the original papers, and we refer the reader to them for details.

For the topic model components of the topic-guided language models, we keep the settings from the original papers. However, in some cases, the original papers use different language model architectures and settings. In order for a topic-guided language model's performance not to be confounded by use of a stronger or weaker language model component, it was necessary to equalize these architectures and settings in the reproducibility study. Specifically, we use 600 hidden units for the language model component and a truncated BPTT length of 30 for all topic-guided language models.

Additionally, we initialize VRTM with pre-trained word embeddings rather than a random initialization, and we strengthen TopicRNN's stop word prediction component by replacing the linear layer with an MLP. We found these changes to help the performance of the respective models, so we included them in the reproducibility study. As noted in the main paper, for rGBN-RNN, we use a model size of 600-600-600 because their public code only supports 3-layer models with same-size RNN layers.

As described in the main text, each model in Table 1 is trained until the validation perplexity does not improve for 5 epochs. After convergence, we use the checkpoint with the best validation perplexity. For each model, we perform three runs with random initializations trained until convergence. We report the mean of these runs along with their standard deviations.

We train LDA via Gibbs sampling using Mallet (McCallum, 2002). The hyperparameters are: $\alpha$ (topic density) = 50, $\beta$ (word density) = 0.01, number of iterations = 1000.

## E   Model Sizes

Table 5 contains parameters counts for the baselines and topic-guided language models. Here, the TGLMs have 100 topics (except rGBN-RNN, which has 100-80-50 hierarchical topics) and the same topic model vocabulary from the APNEWS dataset.

Table 5: Parameter counts for each LSTM-LM baseline and topic-guided language model.

| Model | Excluding embeddings | | | Including embeddings | | |
|---|---|---|---|---|---|---|
| | Total | LM | TM | Total | LM | TM |
| LSTM-LM (sentence-level) | 2.2M | 2.2M | – | 33M | 33M | – |
| LSTM-LM (1 layer) | 2.2M | 2.2M | – | 33M | 33M | – |
| TopicRNN (Dieng et al., 2017) | 2.5M | 2.4M | 0.1M | 45M | 33M | 12M |
| VRTM (Rezaee & Ferraro, 2020) | 3.3M | 2.4M | 0.9M | 37M | 33M | 4.1M |
| LSTM-LM (1 layer, with GRU) | 3.3M | 3.3M | – | 46M | 46M | – |
| TDLM (Lau et al., 2017) | 3.3M | 3.3M | 0.01M | 46M | 34M | 12M |
| LSTM-LM (3 layers) | 7.9M | 7.9M | – | 39M | 39M | – |
| rGBN-RNN (Guo et al., 2020) | 11.6M | 11.6M | 0.05M | 90M | 87M | 3.3M |

## F   Full Probing Results

Table 6 shows the full results for the probe experiment. Standard deviations are computed from three runs with different random seeds for both the LSTM-LM baseline and the topic-guided language model.

Table 6:   Probing results with standard deviations.

| Target | Data | APNEWS | | | IMDB | | |
|---|---|---|---|---|---|---|---|
| | | Acc-1 | Acc-5 | $R^2$ | Acc-1 | Acc-5 | $R^2$ |
| TDLM | init. | .036 (.01) | .135 (.02) | .134 (.00) | .028 (.01) | .132 (.02) | .023 (.00) |
| TDLM | trained | .340 (.01) | .681 (.01) | .621 (.01) | .180 (.02) | .449 (.04) | .400 (.02) |
| TopicRNN | init. | .075 (.00) | .224 (.02) | .022 (.00) | .086 (.01) | .259 (.04) | .014 (.00) |
| TopicRNN | trained | .210 (.00) | .540 (.01) | .232 (.00) | .238 (.02) | .575 (.00) | .231 (.01) |

| Target | Data | BNC | | | WT-2 | | |
|---|---|---|---|---|---|---|---|
| | | Acc-1 | Acc-5 | $R^2$ | Acc-1 | Acc-5 | $R^2$ |
| TDLM | init. | .098 (.02) | .294 (.05) | .073 (.02) | .128 (.05) | .379 (.06) | .018 (.01) |
| TDLM | trained | .314 (.07) | .638 (.07) | .379 (.05) | .510 (.07) | .858 (.02) | .352 (.05) |
| TopicRNN | init. | .077 (.01) | .195 (.05) | .017 (.00) | .197 (.17) | .442 (.10) | -.010 (.00) |
| TopicRNN | trained | .172 (.01) | .424 (.02) | .170 (.00) | .208 (.14) | .486 (.11) | .067 (.01) |

## G   Topic-Guided LM Topics

Table 7 includes randomly sampled topics from each topic-guided language model fit to APNEWS.

Table 7: Randomly selected learned topics from each model on APNEWS.

| Model | Topics |
|---|---|
| TDLM | stolen robber robbed store robbery stole theft jewelry robbers suspect
plane aviation aircraft passengers helicopter pilots airport crashed faa guard
crash driver vehicle truck highway car accident died injuries scene
assange fifa wikileaks nsa snowden blatter iran ukraine journalists russian
emissions nuclear renewable energy congress trade turbines obama reactor china |
| TopicRNN | arriving unsuccessful wash. fail audio bases bargaining sunset first-quarter install
marion evacuate ceiling skull caliber tend evacuation exist shanghai sank
graham turner ellis gordon albany edwards albuquerque davis cia contributions
malloy dannel cheyenne buffalo indian hudson paris india carbon broadway
follow-up scenario rebound rodham luxury rebel ordinary referring prohibiting insist |
| rGBN-RNN | ground site family left dead kilometers miles residents village members
world american disease america military blood days information pentagon top
film art movie actor collection artists artist studio festival theater
recent called lost past washington small big good place today
workers system employees pay cost services agreement authority union contract |
| VRTM | counties family high reported billion prosecutors community percentage asked caliber
angeles vegas moines half ap guilty prosecutors paso smith brown
counties reported high family billion asked prosecutors gov. percentage earlier
high prosecutors gov. family part american recent earlier past long
gov. prosecutors high family recent earlier american past part top |

