# OpenReview forum: "Revisiting Topic-Guided Language Models"
_TMLR — Accepted by TMLR_

### Review · Reviewer_5bYd · 2023-06-20

**Summary Of Contributions:**

This paper compares the effectiveness of topic-guided language models with the standard LSTM language model baselines in a unified setting. Four topic-guided language models and two baselines are evaluated by the held-out predictive performance of each model on three corpora. The authors find that none of these topic-guided language models outperform a standard LSTM language model baseline, and most fail to learn good topics. Probing analysis shows that the baseline’s hidden states already encode topic information, which explains why the topic-guided language models do not yield better performance.

The major contribution is that this work provides an insightful analysis showing that topic-guided language models do not perform effectively as reported in prior work due to inconsistent settings with baselines. In contrast, the LSTM language model has already captured effective topic information inside the hidden states.

**Audience:**

Yes

**Claims And Evidence:**

Yes

**Requested Changes:**

More comprehensive studies with other model architectures and other tasks are suggested.

**Strengths And Weaknesses:**

Strengths:

1. The paper is well-written and easy to read. The problem is clearly defined, the motivation is clear, and the analysis flows smoothly.

2. A comprehensive analysis of topic-guided language models is conducted, reaching insightful conclusions. The conclusion challenges the effectiveness of the methodology in prior work, which would be of interest to the community.

3. This paper would contribute to a line of reproducibility studies that aim to evaluate competing methods in a consistent and equitable manner. The authors promise that the code will be publicly available.

Weaknesses:

1. There are different experimental settings in previous studies. The choice of the unified setting lacks clarity. It is not clear if the conclusion still holds when using different experimental settings, e.g., different or no pre-trained word embeddings, different architecture, etc.

2. Only LSTM approaches are evaluated. As there are transformer-based topic-guided language models proposed in recent years, it would be interesting to see if those transformer approaches also have a similar phenomenon.

---

> ### Author Response · Authors · 2023-07-19
> **Response to 5bYd**
>
> Thank you for calling this paper comprehensive and stating it would be of interest to the TMLR community.
>
> We address your concerns below (additions to the revised paper are colored in blue text):
>
> **Different experimental settings in previous studies**
>
> Thank you for this feedback. In the revised version, we have made the experimental settings described in Appendix D more clear. For the topic model components of the topic-guided language models, we keep the settings from the original papers. However, in some cases, the original papers use different language model architectures and settings. In order for a TGLM’s performance not to be confounded by use of a stronger or weaker language model component, it was necessary to equalize these architectures and settings in the reproducibility study. Specifically, we use 600 hidden units for the language model component and a truncated BPTT length of 30 for all TGLMs.
>
> Additionally, we initialize VRTM with pre-trained word embeddings rather than a random initialization, and we strengthen TopicRNN’s stop word prediction component by replacing the linear layer with an MLP. We found these changes to help the performance of the respective models, so we included them in the reproducibility study.
>
> **Only LSTM approaches are evaluated**
>
> We agree this is important future work and we are also interested in transformer-based TGLMs. However, most of the literature on topic-guided language models has focused on LSTMs, so we followed the literature in studying these models. Our paper demonstrates that existing proposed TGLMs do not outperform LSTM-LM baselines of comparable size and word history conditioning.

---

### Review · Reviewer_5ZBx · 2023-06-28

**Summary Of Contributions:**

The paper takes a deep dive on evaluating work on topic-guided language modeling (TGLM). TGLM aims to improve language model performance by integrating more traditional topic models with RNN-family language models, under the assumption that topic models will capture global document information better than RNNs do on their own. The authors point to multiple flaws in the experimental design of prior work, and show that when these flaws are fixed and thus evaluation is more apples-to-apples, there is no performance gain to TGLM compared to vanilla LMs. The primary results focus on the language modeling performance, but the authors include two sections of additional results, one in which they use probing to measure whether vanilla LMs capture topic information natively, and one in which they look at the quality of the learned topics using automated metrics.


**Audience:**

Yes

**Claims And Evidence:**

No

**Requested Changes:**

Questions about Experiments/Content:
* Since this is a “reproducibility study”, I would like to see more depth and discussion surrounding the differences in your (re)implementation of prior work vs. what was reported in that prior work. In particular, for many models, you say you reimplement from scratch. This is fine—but can you report how your reimplementation compares to the prior work before you make any adjustments to their setup? I.e., can you quantify how much better/worse your implementation’s performance is assuming your goal is just to exactly replicate the prior work? I think it is important to show this, so we can fully understand where the disconnect between your findings and the prior work’s findings stems from. Put differently: a skeptic might say that your lack of difference between models isn’t due to what you say (i.e., isn’t due to a more apples-to-apples comparison) but rather might just mean you did a bad job reimplementing the work! So if you can show that your reimplementation is legit, that strengthens your argument.
* I am skeptical of the section on probing, I think you need to add more in order to make these results compelling. Specifically:
—> Can you include a baseline or two in Table 2? E.g., maybe most frequent topic, and performance of a probe trained on a randomly initialized LM? I don’t know how to contextualize these numbers, so I can’t tell if these support the claim that there is significant topic information captured.
—> Why do you only probe one TGLM? Can you probe the others too, to help contextualize?
—> The issue you mention about how probe performance is correlated with topic model performance concerns me. Can you dig into this more? I feel like there are some artifacts driving the results which aren’t transparently captured in your analysis. Maybe better baselines will help disentangle, or maybe you can run some other analysis to convince us that the probe is capturing exactly what you want it to capture, and not random “other stuff”.
* (This is a thought which is harder to address, so not expecting anything, just dropping it here.) I wonder if you can design a baseline or experiment that gets to the heart of what you are trying to claim here. Namely, I feel like you are trying to say that both topic modeling and language modeling are capturing largely the same stuff, e.g., the principle components of the corpus perhaps? And thus a topic-augmented LM is redundant. Can you quantify this intrinsic connection between the models somehow? Either experimentally or mathematically?

Comments/Suggestions on Writing and Style:
* The paper is nicely written, and whoever did the writing is clearly a good teacher/good at explaining things. :) But that said, there was a lot of time spent on background that was not novel to the paper and could be cut. For example, the section on variational inference felt odd. Either the reader already knows VI and will skip this section, or they don’t know about VI in which case this short description will hardly suffice to catch them up to speed. Moreover, understanding VI is not at all necessary for your contribution. I’d recommend cutting sections of this type of background (on VI, on LSTMs, on LDA, etc) and instead use the extra space to deepen your experimental results and analysis.
* The term “reproducibility study” is a bit of a misnomer here. That implies that you are simply trying to replicate the numbers that others produced, and are not able to do so. But you are actually trying to correct previous problems with experimental design. So it’s not a problem with reproducibility, but rather that when you fix certain problems, you draw an opposite conclusion from the original work.

**Strengths And Weaknesses:**

Overall, the paper represents a nice, if modest, contribution. It is very well written and easy to read. The intuition that the authors support (i.e., that LMs already capture topic information and thus topic modeling is in some ways “redundant”) is an intuition I have personally had when seeing work on TGLMs and so I can’t help but find it satisfying that the authors confirm this intuition. I have some qualms with the experiments (below), in particular the probing studies, and I think augmenting the experiments to address these (or rearranging so that material from appendix can appear in the main text, if needed) would significantly strengthen the work.

---

> ### Author Response · Authors · 2023-07-19
> **Response to 5ZBx**
>
> Thank you for stating this paper is well written and that it confirms your previously held intuitions on TGLMs.
>
> We conduct the requested experiments and address the additional concerns below (additions to the revised paper are colored in blue text):
>
> **Differences in re-implementation of prior work vs. what was reported in prior work**
>
> Thank you for your suggestions. We implemented the suggested experiments and report results below. We attempted to exactly replicate the prior work in our codebase -- the table includes the TGLM and the prior work’s corresponding LSTM-LM baseline, respectively, in alternating rows (except for TopicRNN, which only reports an RNN baseline; we still train an LSTM for our corresponding baseline and for TopicRNN). For datasets, TDLM and VRTM use APNEWS, and TopicRNN uses PennTreeBank.
>
> |  | Our implementation | Prior work |
> |-|-|-|
> |TDLM | 48.3 | 52.8 |
> | | 54.5 | 64.1 |
> | VRTM | 41.7 | 47.8 |
> | | 38.4 | 62.8 (from previous work) |
> | TopicRNN | 123.7 | 99.5 |
> | | 116.6 | 124.7 |
>
> TDLM: We are able to replicate TDLM’s improvement over a sentence-level LSTM-LM baseline.
>
> VRTM: We use the vocabulary size reported in VRTM’s Appendix D, 7.8K words. However, this vocabulary size is inconsistent with the baseline model from previous work reported in the paper, which uses 34K words (we discuss this in Section 6). Perplexity is tied to vocabulary size, with smaller vocabularies leading to improved perplexities. In VRTM’s codebase, they also only train on the first truncated BPTT sequence in the document (45 words).
>
> TopicRNN: We are unable to replicate TopicRNN performance on PennTreeBank; we find no improvement over a standard LSTM-LM. We checked our model implementation for correctness and cannot determine the cause of this discrepancy, as TopicRNN has no public codebase.
>
> **Including a probing baseline**
>
> Thanks for this suggestion; we added the probing result for a randomly initialized LSTM-LM to the revised Table 2.
>
> We also tested a second baseline where we do not train a probe, and instead predict the mean transformed topic vector over the training data (for top-1 accuracy, this is equivalent to predicting the most frequent topic). The performance for both baselines were very similar – this is expected, since if the LSTM-LM hidden states contain no information about the topic vector (as is the case for random initialization), the MSE loss minimizer is to predict the mean of the training data.
>
> **Probing other TGLMs**
>
> In the paper, we stated that we chose TDLM for the probing task because it had the highest quality topics according to coherence (Table 3). Our intuition was that “better” topics would be more helpful to language modeling and thus captured in the LSTM-LM hidden representations to a greater extent. To demonstrate, we also run the probe task and a baseline for TopicRNN, a TGLM with lower quality topics, and report the results in the revised Table 2. TopicRNN’s topic vector is captured by the LSTM-LM, although to a lesser extent than TDLM’s, supporting our intuition.
>
> **Concern that probe performance is correlated with topic model performance**
>
> When we stated this held for TDLM, our intuition was that better topics are more useful to language modeling and will be captured more in the representations. We believe that the additional baselines and probing for TopicRNN’s document-topic vectors helps contextualize the probing results better.
>
> **Quantifying connection between topic modeling and language modeling**
>
> This is an interesting perspective – to rephrase, both language models and topic models are performing representation learning on documents, and the goal is to characterize the common features that are captured by both models. Topic models which factorize the term-document matrix are capturing something like principal components as topics (e.g., LSI performs SVD on the term-document matrix). Maybe a relevant experiment could be probing the language model for the representations learned by topic models like LDA.
>
> **Cutting sections of background information**
>
> Thanks for this feedback. In addition to the reproducibility experiments, we also wanted the paper to stand alone as much as possible in describing TGLMs as a model class – hence including short reviews on VI, LSTMs, and LDA. In addition to the short review, the VI section introduces the ELBO optimization objective and notation. We have also expanded the probing analysis according to your suggestions.
>
> **Reproducibility study is a bit of a misnomer**
>
> Our work focuses on conceptual, rather than direct, replication. We test the same claim made in the TGLM literature – that these models outperform vanilla LSTM-LMs – and use the same models, but correct the experimental design. For TopicRNN, we are also unable to replicate the results using the same setup as described in their paper, but we can’t determine whether the discrepancy is also due to experimental design because of the lack of a public codebase.

---

### Review · Reviewer_dbWw · 2023-07-05

**Summary Of Contributions:**

This paper revisits __topic-guided language models__, broadly defined as language models that combine the standard causal, left-to-right language modelling next-word prediction task with __topic models__ (Blei et al., 2003; inter alia). This line of work is motivated by earlier results on the limited ability of RNN and LSTM-based models in capturing long-range dependencies. In theory, such combination of language models and topic models would enhance topic-guided language models' ability to model long-range dependencies, by decoupling the predictions that only require *local* context (e.g. local syntactic and semantic dependencies), which RNN and LSTM-based language models can already do quite well, from those that require long-range, *document-level*, semantic context, which topic models are able to capture well by uncovering latent topics. To that end, prior work has claimed positive results for topic-guided language modelling, demonstrating that such topic-guided language models achieve better perplexity than standard LSTM-based models, whilst also learning interpretable topics as an important byproduct; this stands in contrast to the hidden state activations of standard LSTM-based language models, which are comparatively harder to interpret.

The paper begins with a review of both language models (in particular, those that are based on RNNs and LSTMs) and topic models, before summarising four major prior work in topic-guided language modelling, covering both __topic-biased language models__ and __joint topic and language model__. The paper then asks three key questions:
- Under comparable and scientifically rigorous experimental conditions, to what extent --- if at all --- do topic-guided language models really outperform the standard LSTM language model baselines that are properly tuned to the same extent? In this work, the comparable experimental condition is achieved by controlling for three factors: (i) First, the topic model component should depend *only on the preceding words*, in order to preserve the left-to-right causality of the language modelling task; (ii) second, the baseline LSTM models should condition on prior words in the *entire document* history (as is standard in the language modelling literature), as opposed to only the prior words in the current sentence, as done in most prior topic-guided language modelling work; and lastly, (iii) the baseline models and the topic-guided ones should be compared under *similar model sizes*, as model size can be an important determinant of language modelling performance.
- To what extent --- if at all --- do the learnt topics from topic-guided language models outperform those much simpler, standard topic model approaches?
- To what extent can a properly tuned, standard LSTM-based language model encode a similar degree of topic information (in its hidden state activations) as the latent topics uncovered by more complicated, topic-guided language modelling approaches?

To that end, the paper found that, under comparable and scientifically rigorous experimental conditions as outlined above, the baseline LSTM LM approach can actually __outperform__ the four topic-guided language models in terms of perplexity; these results are consistent across three different document-level language modelling benchmarks. Furthermore, the latent topics uncovered by these topic-guided language models are, in fact, __less coherent__ than those found by a much simpler, standard LDA (Blei et al., 2003) approach. Finally, despite the baseline model's lack of an explicit objective function or encouragement to uncover latent topics, probing analysis demonstrates that the hidden state activations of standard LSTM LMs do, in fact, encode a fairly similar extent of latent topic information as that uncovered by a topic-guided language model approach (TDLM, Lau et al., 2017).

**Audience:**

Yes

**Broader Impact Concerns:**

I do not foresee any broader impact concerns.

**Claims And Evidence:**

Yes

**Requested Changes:**

- **Strongly Recommended**:  Conducting experiments on a more commonly used document-level language modelling benchmark, such as WikiText 2 or WikiText-103. This would give the readers a better idea of how the numbers compare with strong language models in prior work (weakness 1 above).
- **Strongly Recommended**: Improving the mathematical notations to be more standard and precise (weakness 2 above).
- **Strongly Recommended**: Re-running probing analysis with either control task or minimum description length probing, to assess how much of the probe results are really due to the probed information being present in the hidden state activation (as opposed to the classifier's expressivity) (weakness 3 above).
- **Strongly Recommended**: Resolving the questions (and incorporating the relevant reply into the paper where applicable), stylistic suggestions, and citation as raised above.

**Strengths And Weaknesses:**

__Strengths__
- Given the incredibly fast pace of the machine learning, NLP, and language modelling literature, __reproducibility and scientific rigor__ --- in particular, by conducting fair comparisons of different models under comparable experimental conditions, and accounting for various potential confounders like model sizes, proper tuning of the baseline models, etc. --- are extremely important in the field. This paper makes an important step in this direction: By comparing standard, well-tuned LSTM language models with their topic-guided counterparts in a level playing field, the paper derives useful insights (primarily around the fact that topic-guided language models do *not* necessarily outperform standard, well-tuned LSTM baselines, and that they do not necessarily learn coherent topics) that will be useful for other researchers in the field and the broader community. This kind of work is important to advance our __scientific understanding__ of what works well today (and what doesn't), above and beyond achieving the next state of the art results.
- The paper provides a comprehensive review of the important concepts, such as topic models and topic-guided language models, that are necessary for less familiar readers to understand the key ideas, and therefore interpret the paper's key findings. I find the review of background material to be particularly thorough, especially given the page limit.
- The paper conducts experiments on three different document-level language modelling datasets, and also takes into account the variance resulting from different random seeds, which improves the credibility of the findings and reflects more scientific rigor.

__Weaknesses__
- The choice of evaluation datasets should be broadened to include more commonly used document-level language modelling benchmarks, such as WikiText-2 and WikiText-103. This would make it easier to benchmark the paper's results against prior work on strong, document-level language models, which is difficult to do at the moment (currently Table 1 in the paper only includes the authors' own LSTM-LM implementation as the baseline). This would enable the readers to assess how the authors' baseline implementation compares against strong LSTM LMs used in prior work (e.g. Mogrifier LSTM, Melis et al., 2019), including those with a similar number of parameters. This would further improve the credibility of the findings.
- The mathematical notation in the paper can be improved to be clearer and more precise. Following standard notation, I would recommend using bold uppercase letters for matrices, bold lowercase letters for vectors, and non-bold letters for scalars. Similarly, I would recommend using bold lowercase letters to denote a __sequence of words__ (e.g. $\mathbf{x}_{<t}$ ). In contrast, a single word can be denoted with a lower, non-bold, italic lowercase letter, such as $x_t$.
- The probing analysis --- while helpful --- can be strengthened by using control task (Hewitt and Liang, 2019) or minimum description length (Voita and Titov, 2020). This would help disentangle how much of the probing performance is due to the presence of the information in the hidden state vector, as opposed to the strength of the classifier itself.
- Because the primary benefit of topic-guided language models is claimed to be long-range, document-level semantic coherence (maybe human evaluation over the generated text?), it would be nice if there is a targeted metric for this, to assess whether or not such models really have an advantage over standard, well-tuned LSTM LMs in terms of long-range semantic coherence.
- There are still some open questions, suggestions, and stylistic suggestions (as listed below) that are not resolved yet.

__Questions__
- Does the topic-biased language model (section 4.1) basically learn to interpolate between the topic model and the standard LSTM LM? If so, it might be worth saying more explicitly.
- In the "Inference" section of the TDLM, the topic model (TM) and the language model (LM) have two different sets of parameters. But in Eq. (7), $\hat{\boldsymbol{\theta}}$ is based on an encoding of the prefix by the topic model (TM) parameters $\boldsymbol{\theta}^{\text{TM}}$ , which is then fed into the LM parameters, even though the equation above it mentioned that $\ell_{\text{LM}}$ depends on $\boldsymbol{\theta}^{\text{LM}}$, not $\boldsymbol{\theta}^{\text{TM}}$. Could you please clarify this?
- The backpropagation through time used a truncated length of 30 (page 8, training details). This seems to be on the shorter end; did you try how the results would change with a longer truncated length?
- Another potential benefit of topic-guided language models is the __controllability__ angle, as we can ask the model to generate an article for a given topic vector. Having a discussion on whether this advantage is likely to hold (or mentioned as a potential topic of investigation for future work) would be nice.

__Stylistic Suggestions__
- In Table 1, I recommend putting the best entry model (lowest ppl.) for a given parameter count in bold, which would make the table much easier to read (this might require some rearranging of how the table is structured).
-  In page 3, "memoizing" is a typo.
- In page 9, "prevelant" is a typo.
- In page 10, "the the" is a grammatical error.
- I find the last sentence in the conclusion ("Future work can prioritize...") to be a bit hard to parse.

__Citation__
There is a recent paper that discusses a similar issue (lack of fair comparison and scientific rigor when comparing different models, leading the community to have the wrong understanding of what works and what doesn't): Nityasya et al. (2023): "On "Scientific Debt" in NLP: A Case for More Rigour in Language Model Pre-Training Research".

---

> ### Author Response · Authors · 2023-07-19
> **Response to dbWw**
>
> Thank you for an especially thorough summary of this paper’s contributions, and for stating that the reproducibility and scientific rigor our paper aims for is important to the field.
>
> We address your concerns and questions below (additions to the revised paper are colored in blue text):
>
> **Conducting experiments on a more commonly used document-level language modeling benchmark**
>
> The aim of our paper is to show that topic guidance from existing TGLMs do not improve language modeling performance over a vanilla LSTM-LM – one without modifications made in e.g., Mogrifier LSTM (Melis et al., 2019) that further strengthens the language model. These strong/SOTA LSTM-based language models also have up to an order of magnitude more parameters and would not be directly comparable to the smaller baselines we study in this paper.
>
> **Improving mathematical notation to be more standard and precise**
>
> Thanks for pointing this out; we have incorporated the suggestions in the revised version.
>
> **Re-running probing analysis with either control task or MDL probing**
>
> We implemented a control task according to the recipe provided in Hewitt and Liang (2019): “Designing and Interpreting Probes with Control Tasks.” First, we map each word in the vocabulary to a fixed but randomly chosen topic. The probe task is to predict the TGLM document-topic vector from the LSTM-LM hidden states. To construct control task examples, given an LSTM-LM hidden state at time $t$ for document $d$, we take the corresponding word at time $t$ for document $d$ and determine its topic according to the control task map. With $K$ as the number of topics, we then set the document-topic vector to be $0.9$ at this topic and $0.1 / (K-1)$ at the remaining $K-1$ topics.
>
> We found that the linear probe we used in the paper was unable to learn the control task (the probe overfits and doesn’t generalize to the test data); we also tried a 1-layer MLP probe, which had the same issue. This may be because our language model has less parameters and is trained on less data compared to the model used in Hewitt and Liang – they use ELMo, which has ~94M parameters, while our LSTM-LM only has 2.2M parameters – and due to the hidden representation being lower quality compared to ELMo’s, the probe cannot reconstruct the exact word at time $t$ that produced it. If the hidden representation doesn’t contain enough information to learn the control task, probe selectivity is arguably less of a concern.
>
> **Resolving questions, stylistic suggestions, citation**
>
> We address the questions individually below. Thanks for the stylistic and citation suggestions – we have incorporated them in the revised version. Regarding restructuring Table 1, we would prefer to keep the current format without bolding specific entries. Our rationale is that Table 1 is meant to provide an overview of all models rather than draw attention to particular entries.
>
> **Does topic-biased language model learn to interpolate**
>
> That’s correct. Thanks for the suggestion; we have stated this in the revised version.
>
> **Clarifying TDLM inference section**
>
> $\boldsymbol\theta$ is the output of the encoder function, enc(.), which takes a bag-of-words as input. During training, the encoder receives gradients from both the topic modeling and language modeling objectives – however, the encoder receives different bag-of-words inputs for each (the language model bag-of-words input excludes the current sequence being predicted), hence we call the encoder’s output for the topic modeling objective $\boldsymbol\theta^{TM}$ and its output for the language modeling objective $\boldsymbol\theta^{LM}$. During inference, the input to the encoder is restricted to only the previous topic words in the document, and we call the output of the encoder during inference $\hat{\boldsymbol\theta}$.
>
> **Trying a longer truncated BPTT length**
>
> We tried increasing the length in our initial experiments, but due to the smaller capacity of these language models in terms of parameter count, perplexity did not improve by much.
>
> **Discussing controllability as another potential benefit of TGLMs**
>
> Thanks for this suggestion – we have mentioned this as a subject of future work in Section 7 in the revised version.

---

> > ### Comment · Reviewer_dbWw · 2023-08-04
> > **Thank You for the Response**
> >
> > Thank you for the response. Having also read the other reviews and the authors' response to them, I think this paper would be a good contribution for the TMLR audience. I believe that, as a community, we should conduct more work like this that critically reexamines prior models and claims under scientifically rigorous, comparable experimental conditions where the (strong) baseline and the alternative approaches are tuned to the same extent.
> >
> > On this note, however, I still think it will be helpful to run experiments on more commonly used, document-level language modelling datasets. This would at least be useful to ascertain that the authors' LSTM implementation is *comparable* to prior LSTM baselines (i.e. the vanilla version, not with Mogrifiers, etc.) with similar model sizes. I observe that Reviewer 5ZBx also has a similar (albeit differently-worded) point requesting more details regarding the authors' reimplementation vs prior work. To that end, having results on more commonly used document-level LM datasets would address this point, and further improve the credibility of the findings.
> >
> > Also, the fact that the control task probe does not overfit, even with a 1-layer MLP, makes me curious how the findings will change when the authors use larger models (along with the latest bells-and-whistles, such as layer normalisation, residual connections, etc.). Using a larger model size would also put the authors' implementation more comparable to that of prior work on strong LSTM baselines, and understand whether larger models learn better topics or diverge more (or less) from topic-guided LMs, which would be an interesting question to explore on its own.
> >
> > Overall, I think the paper is strong and useful enough without these additions, but it might be something to consider for the final version.

---

### Author Response · Authors · 2023-11-03
**Uploaded the camera-ready version**

Dear reviewers and editorial team,

Thank you for your insightful comments and discussion, which have helped to greatly improve the paper. We incorporated the following reviewer suggestions in the camera ready version:

* Broadened evaluation that includes a more commonly used document-level language modeling benchmark dataset, WikiText-2. Results on this dataset have been added to the experiments section, which reaffirm the original findings.
* Probe experiments with another topic-guided language model, TopicRNN, and initialization-only baselines.
* Stylistic changes and clarifications to improve presentation and readability.

---

### Decision · Action_Editors · 2023-08-07

**Recommendation:** Accept with minor revision

**Comment:**

Overall, it is a good study and makes good contributions to the community (more details can be found in reviews).

My recommendation is "accept with minor revision".

The authors need to address the concern about test datasets and revise the paper:
This work focuses on empirical study, but conducted experiments on less well-used datasets, which makes the conclusions less convincing, as there is not much prior work on these datasets. Therefore, the authors need to test on widely used benchmarks like WikiText-103 or other larger datasets and add new results into the final version.

**Audience:**

Yes

**Claims And Evidence:**

Yes. But it is better to test on widely used benchmarks like WikiText-103.